# A Multi-Channel Ensemble Method for Error-Related Potential Classification Using 2D EEG Images

**DOI:** 10.3390/s23052863

**Published:** 2023-03-06

**Authors:** Tangfei Tao, Yuxiang Gao, Yaguang Jia, Ruiquan Chen, Ping Li, Guanghua Xu

**Affiliations:** 1Key Laboratory of Education Ministry for Modern Design & Rotor-Bearing System, Xi’an Jiaotong University, Xi’an 710049, China; 2School of Mechanical Engineering, Xi’an Jiaotong University, Xi’an 710049, China; 3School of Foreign Studies, Xi’an Jiaotong University, Xi’an 710049, China; 4State Key Laboratory for Manufacturing Systems Engineering, Xi’an Jiaotong University, Xi’an 710049, China

**Keywords:** attention-based convolutional neural network, brain–computer interface (BCI), error-related potential (ErrP), multi-channel ensemble

## Abstract

An error-related potential (ErrP) occurs when people’s expectations are not consistent with the actual outcome. Accurately detecting ErrP when a human interacts with a BCI is the key to improving these BCI systems. In this paper, we propose a multi-channel method for error-related potential detection using a 2D convolutional neural network. Multiple channel classifiers are integrated to make final decisions. Specifically, every 1D EEG signal from the anterior cingulate cortex (ACC) is transformed into a 2D waveform image; then, a model named attention-based convolutional neural network (AT-CNN) is proposed to classify it. In addition, we propose a multi-channel ensemble approach to effectively integrate the decisions of each channel classifier. Our proposed ensemble approach can learn the nonlinear relationship between each channel and the label, which obtains 5.27% higher accuracy than the majority voting ensemble approach. We conduct a new experiment and validate our proposed method on a Monitoring Error-Related Potential dataset and our dataset. With the method proposed in this paper, the accuracy, sensitivity and specificity were 86.46%, 72.46% and 90.17%, respectively. The result shows that the AT-CNNs-2D proposed in this paper can effectively improve the accuracy of ErrP classification, and provides new ideas for the study of classification of ErrP brain–computer interfaces.

## 1. Introduction

A brain–computer interface (BCI) is a system that does not rely on peripheral nerves and can control external devices by directly decoding the users’ neuronal activity [1]. It can help patients with limb movement difficulties to communicate with the outside world, as well as help stroke patients restore motor function [2,3,4].

Electroencephalography (EEG) is currently the most commonly used method for BCI systems to acquire signals. With the advantages of easy collection, high temporal resolution, low cost, and non-invasiveness, EEG measures weak electrical potentials in the brain through channels placed on the scalp [5].

However, due to some limitations of EEG signal, such as non-stationarity, nonlinearity and low signal-to-noise ratio, a large amount of standardized data is difficult to obtain. Therefore, the decoder usually misidentifies the intention of the subject, greatly limiting the practical application of EEG-based BCI [6,7,8].

An error-related potential (ErrP) is a kind of event-related potential (ERP). When the expectations of the subjects are inconsistent with the actual results [9] error-related potential (ErrP) will be generated in the anterior cingulate cortex (ACC) [10]. The waveform characteristics can be observed on average through multiple trails. A positive peak appears about 200 ms after the feedback is presented, a large negative peak appears about 250 ms and a positive peak appears about 320 ms [3,11]. Due to the different experimental paradigm, the waveform characteristics will be slightly different. The waveform of channel FCz obtained in this paper is shown in Figure 1.

Studies have shown that ErrP is an inherent feedback mechanism of humans and can generate in the brain without training [9]. What is more, the signal waveform will not change significantly and the performance of the corresponding classifier will not reduce greatly after a long period of time [12]. Additionally, some studies have found that ErrP can be detected in a single trial [9], which means that ErrP can be combined with other BCIs to correct a BCI’s erroneous instructions, thus improving the BCI’s overall performance [13,14,15].

In order to use ErrP to correct the BCI system instructions and improve the overall performance of the system, a high-accuracy ErrP recognition is necessary. Therefore, the accurate detection of ErrP from the EEG signal is crucial to improve the performance of this type of BCI system. Much research has been carried out to classify ErrP signals.

There exist two categories of ErrP classification methods. One is based on a traditional machine learning method, manually selecting the features of ErrP and then training a classifier. Another can automatically extract and classify signal features using a convolutional neural network based on the deep learning method. 

Ricardo and Millan et al. [16] chose two channels (FCz and Cz) and input them into the Gaussian classifier for classification. Channels (FCz, Cz, or both) and time windows used for classification were selected independently per subject based on the classification performance. 

Akshay Kumar et al. [17] subdivided each epoch of the data into 13 time windows and calculated the average of the mean value of each time window. These mean values were used as a feature set to train the linear discriminant analysis (LDA) classifier.

Praveen K et al. [18] took an electrode-averaged feature extracted from trials as a classification feature and then adopted an LDA classifier to classify correct and incorrect events.

The above traditional machine learning methods have made some achievements by manually selecting the characteristic periods, frequency bands and channels of ErrP. However, part of the effective feature information may be ignored by manually extracting features. In addition, the generalization ability of the algorithm may be reduced, for the performance depends on manually tuning parameters. To improve the accuracy and generality of the ErrP classification algorithm, more researchers have focused on deep learning methods that can automatically extract effective features in recent years.

Mayor Torres et al. [19] designed a new convolutional neural network named ConvNet and proposed the inclusion of error-related potential activity as a generalized two-dimensional feature set. They have made some performance improvements.

Akshay Kumar et al. [20] adopted a convolutional neural network-based double transfer learning method. The first stage aims to capture the global features of ErrP. In the second stage, each subject is individually tuned to transfer from global features to local features. This method outperformed the existing statistical classifier methods.

Nayab Usama et al. [21] used a multi-layer perceptron artificial neural network (MLP ANN) to classify error and correct responses. They considered different brain regions and the results showed that the frontal and central regions were the most significant.

These methods can learn the signal features automatically based on deep learning. This kind of algorithm generally outperforms the traditional machine learning methods. Although it embodies the great potential of the deep learning method in the ErrP processing algorithm, it still cannot satisfy the demands in actual application. Current deep learning has entered into the commercial stage in image classification tasks [22]. Some researchers convert 1D electrocardiogram (ECG) signals into 2D images and then input them to 2D CNN, showing better performance than 1D CNN [23,24]. Similar to the periodic electrocardiogram, ErrP can recognize the peak information on the feature period with naked eyes. This demonstrates that the deep learning-based image processing methods have great potential for ErrP classification and our earlier research has also confirmed it [25]. In addition, the current research mainly focuses on the selection of channels FCZ and CZ, and the optimization of channel selection has received little attention. In fact, ErrP features also exist in other areas of the anterior cingulate cortex, which is also conducive to the final classification. However, due to the different signal characteristics of different channels, the final classification contribution is different. How to integrate the information of each channel effectively is also the key to improving the performance of the model. As a common integration method, a majority voting algorithm is widely used, but its combination strategy is too simple to highlight the “more valuable” channel with all the channel classifiers having the same weight. Deep learning has become a powerful method of learning, capable of automatically learning complex structures from data [26,27]. Therefore, we adopt deep learning to learn the complex relationships between different channel classifiers and tags, that is, assigning different weighting to different channel classifiers.

In this paper, we propose a multi-channel ensemble method for ErrP detection using 2D EEG images. We input the ErrP time-domain signal into the convolutional neural network in the form of a 2D grayscale image. Then, we design the network architecture on the grounds of signal characteristics of ErrP, and fuse the feature information of multiple channels at the decision-making end. The advantages of the proposed ErrP detection method are validated on the Monitoring Error-Related Potential dataset and the results show that our proposed method outperforms other methods. The remainder of this paper is organized as follows: the multi-channel ensemble method ErrP classification proposed in this paper is introduced in detail in Section 2. The details of experiments are presented in Section 3. Section 4 discusses the experimental results. Finally, the conclusion is presented in Section 5.

## 2. Methods

The anterior cingulate cortex (ACC) is believed to be the brain region responsible for the generating of ErrP, and ErrP waveform characteristics can be clearly observed on the channels FCZ and CZ in the ACC [28,29,30]. Most researchers [19,31,32] usually manually select the signals of these two channels and input them into the neural network in the form of 1D data for classification. However, EEG signals usually contain a lot of noise, which makes it difficult for neural networks to learn robust features, and manual selection of these two channels will miss the feature information of other channels.

In this paper, we proposed a multi-channel ensemble method for ErrP detection using 2D EEG images. Figure 2 illustrates the overall architecture of our model. First, we selected several channels from the anterior cingulate cortex (ACC) containing ErrP signatures. Then, the 1D EEG signal of the selected channel was preprocessed and converted to 2D EEG image, and input into the AT-CNN model, respectively. Each AT-CNN was classified to provide local decision information for that channel. A channel ensemble classifier was used to integrate the local decisions of each channel to make the final decision. Since the signal characteristics of each channel are different, each AT-CNN model needs the EEG data of the corresponding channel for training. After the training, each model plays the role of an expert.

The core component of the system is an AT-CNN model, which is proposed to fully extract the spatiotemporal features of a 2D EEG image. We will introduce this deep model in detail later.

### 2.1. Channel Selection

The ErrP signal feature appears in the ACC, and the ErrP feature can be observed in multiple channels of the anterior cingulum belt layer. Obvious waveform features in the channel FCZ can be observed, and the ErrP waveform feature around the channel FCZ gradually weakens [12]. Selecting too many channels not only increases noise but also the model inference time. Thus, selecting the appropriate channel can effectively utilize the discrimination information provided by each channel and improve the ErrP classification performance. In Section 3, we will discuss the effect of different channel group selection on ErrP detection accuracy.

### 2.2. Data Preprocess

The raw EEG signal contains a lot of noise. To improve the signal-to-noise ratio, the raw EEG signal is spatially filtered using common average reference (CAR) and then filtered to [1,2,3,4,5,6,7,8,9,10,11] Hz using a third-order Butterworth band-pass filter. To eliminate eye artifacts and other low-frequency artifacts, we use ICA for artifact elimination. After the above operation, we use max-min normalization to normalize the signal and transform EEG signals into EEG images by plotting each EEG signal as an individual 224 × 224 grayscale image including 0.2 s before stimulus and 0.8 s after stimulus, for a total of 1 s. Cropping the first 0.2 s can make the neural network aware of the changes in the onset and offset of EEG signal stimulus presentation, which is useful information for CNN. The plotting operations are implemented using the Matplotlib library in Python. As shown in Figure 3, we assume that the lower left corner of the image is the reference coordinate (0,0) and the origin of the coordinate is located at the position of coordinate (0,124). The x axis is the direction of the time axis, and the y axis is the direction of the amplitude. When we input the neural network, we hide the x and y axes. The pixel of the waveform line is 0, and the pixel of the remaining blank area is 255.

### 2.3. Attention-Based Convolutional Neural Network

#### 2.3.1. Architecture

A schematic diagram of the AT-CNN structure is shown in Figure 4. The architecture of AT-CNN is shown in Table 1. A module is repeated four times, each with a convolution layer, ReLU activation function, batch normalization and pooling layer, respectively. A convolutional block attention module (CBAM) is added after the above operation, which can learn what and where to emphasize or suppress to improve the performance of the CNN [33]. The fully connected layers are used at the end of the above layers to obtain the final classification result. Dropout is used between the fully connected layers to avoid overfitting.

The function of the convolution layer is to extract features from images. The operation of the convolutional layer is defined in Equation (1):(1)$$x_{j}^{l}=f\left(b_{j}^{l}+\sum_{i\in M_{j}}w_{ij}^{l}*x_{i}^{l-1}\right)$$
where $x_{j}^{l}$ represents the jth feature map of the lth layer, *f* denotes the activation function. ReLU is used in our network, $M_{j}$ denotes the set of input feature map, $w_{ij}^{l}$ is the convolution kernel and $b_{j}^{l}$ is bias. 

The batch normalization (BN) layer [34] is followed by each convolutional layer. BN can effectively alleviate overfitting and make the network training process more stable. BN is applied to each mini-batch, and the formula for mini-batch normalization is as follows:(2)$${\hat{x}}_{i}^{\left(k\right)}=\frac{x_{i}^{\left(k\right)}-{µ}_{B}^{\left(k\right)}}{\sqrt{{ó}_{B}^{{\left(k\right)}^{2}}+\varepsilon }}*\gamma +\beta$$
where $x_{i}^{\left(k\right)}$ is the ith element of the *k*-th dimension, ${µ}_{B}^{\left(k\right)}$ and ${ó}_{B}^{{\left(k\right)}^{2}}$ are the mean and variance of the input mini-batch, respectively, and γ and β are hyperparameters that are trained to adjust the mean and deviation of normalized data.

The BN layer is followed by the max pooling layer. The main goal is to preserve the main features while reducing the parameters. The maximum pool operation is used, defined as follows:(3)$$x_{j}^{l}=\max\left(x_{j}^{l-1}\right),l\in P_{j}$$
where $P_{j}$ is the set of input feature map.

CBAM includes two independent sub-modules, a channel attention module (channel attention module, CAM) and a spatial attention module (spatial attention module, SAM), which can perform attention in channel and space dimensions. These two modules are connected sequentially. Figure 4 depicts the calculation process of each attentional map. The overall attention process can be summarized as:(4)$$F^{c}=M_{c}\left(F\right)⊗F$$
(5)$$\;F^{S}=M_{s}\left(F^{c}\right)⊗F^{c}$$
where ⊗ denotes element-wise multiplication and F denotes input feature map. 

CAM focuses on ‘what’ is significant given an input image, and the channel attention is computed as:(6)$$M_{c}\left(F\right)=ó\left(M_{1}\left(M_{0}\left(F_{avg}^{c}\right)\right)+M_{1}\left(M_{0}\left(F_{max}^{c}\right)\right)\right)$$
where ó denotes the sigmoid function, $M_{0}\in R^{C/r\times C}\;\mathrm{and}\;M_{1}\in R^{C\times C/r}M_{0}$ and $M_{1}$ are MLP weights that shared for both inputs and the activation function is ReLu. $F_{avg}^{c}$ and $F_{max}^{c}$ represents the global average-pooled features and the global max-pooled features that contain the spatial information of a feature map.

Unlike CAM, the SAM is focused on “where” is an information part. Spatial attention was calculated as follows:(7)$$M_{s}\left(F\right)=ó(f^{7\times 7}\left(\left[F_{avg}^{s};F_{avg}^{s}\right]\right)$$
where ó denotes the sigmoid function, $f^{7\times 7}$ denotes a convolution operation with the kernel size of 7 × 7. $F_{avg}^{s}$ and $F_{max}^{s}$ represents average-pooled features and max-pooled features across the channel that contain the channel information of a feature map.

The CBAM will benefit the model on two levels. First, CAM can help the model focus on the rich and important part of the 2D EEG image, thus improving the detection performance. Secondly, SAM, as a supplement to CAM, can help the model highlight the area with the largest amount of information in the 2D EEG image.

The full connection layer is located behind the convolution layer and plays the role of classifier in the whole convolutional neural network. Before the input of the FC layer, the output characteristics of the previous layer are tiled into a one-dimensional vector form. After the full connection layer, the prediction results are obtained. For the work in this paper, two nodes are output, which respectively indicate that ErrP is detected and ErrP is not detected from the EEG signal.

#### 2.3.2. Loss Function

The loss function is used to calculate the difference between the predicted value and the true value. The smaller the loss function, the better the model fits. The cross-entropy loss is commonly used in binary classification. In this paper, we employ label smoothing [35]. As shown in Equation (4), C is the loss function that the optimizer uses to minimize it, N is the number of all samples, $p_{c}$ is the output value of the neural network (between 0 and 1) and $y_{c}^{\prime }$ is the target label. Here, as shown in Equation (5), instead of using the original target label $y_{c}$, we adjust the actual label $y_{c}$ to $y_{c}^{\prime }$ and set *ε* to 0.1 (replace the exact classification target from 0 and 1 to *ε* and 1−*ε*). Label smoothing can prevent the model from pursuing the probability of accuracy and improve the generalization performance of the model.
(8)$$C=-\frac{1}{N}\sum_{i=1}^{N}\left[y_{c}^{\prime }Logp_{c}+\left(1-y_{c}^{\prime }\right)\log\left(1-p_{c}\right)\right]$$
(9)$$y_{c}^{\prime }=\left(1-\varepsilon \right)*y_{c}$$

#### 2.3.3. Detailed Implantation

To meet the requirements of the experimental setup, the obtained ErrP dataset is often unbalanced, such as the dataset with only 20% erroneous trials, which is introduced in Section 3. To balance the dataset, a common approach is random undersampling, randomly eliminating the majority class of data to balance the dataset, which may discard a lot of useful information. Another common approach is random oversampling, which randomly extracts samples from the minority class to replicate. However, this may lead to overfitting. Studies have shown that random oversampling is more advantageous than undersampling [36]. Therefore, we adopt random oversampling in this paper and the minority class is randomly copied to the same as majority class.

During model training, we set 100 epochs in advance, and the initial learning rate is 0.001. We use cosine learning rate decay to dynamically reduce the learning rate, and the optimizer uses Adam [37]. In addition, we also use early stopping [38]. If no decrease in validation loss is observed within 20 epochs, the model training is terminated to avoid overfitting. This model is implemented in Pytorch.

#### 2.3.4. Performance Metrics

For evaluation, we refer to the EEG signal with ErrP as a positive sample, and the EEG signal without ErrP as a negative sample. We use accuracy, sensitivity and specificity to evaluate the system. These performance metrics are as follows:(10)$$\mathrm{Accuracy}=\frac{\mathrm{TP}+\mathrm{TN}}{\mathrm{TP}+\mathrm{FP}+\mathrm{FN}+\mathrm{TN}}$$
(11)$$\mathrm{Sepecifity}=\frac{\mathrm{TN}}{\mathrm{TN}+\mathrm{FP}}$$
(12)$$\mathrm{Sensitivity}=\frac{\mathrm{TP}}{\mathrm{FN}+\mathrm{TP}}$$
where TP, FN, TN and FP represent the number of true positive, false negative, true negative and false positive ErrP trials, respectively.

### 2.4. Multi-Channel Ensemble Approach

Some channels, such as FCZ or CZ, are available for ErrP detection. However, none of these are fully accurate and classifiers may make mistakes under some circumstances. Stacking of multiple different AT-CNN models will lead to performance improvement over individual models.

Figure 5 shows our multi-multi-channel ensemble method. In our method, we divide the multi-multi-channel ensemble model into two stages. one for providing the local decision information of each channel, and the second stage for integrating the local decision information of the first stage and making the final decision. 

The training process of AT-CNN of each channel and ensemble classifier is divided into two stages as shown in Figure 5. In the first stage, we divide the dataset into five subsets of the same size, $\left\{A_{1},A_{2},A_{3},A_{4},A_{5}\right\}$,where $A_{i}=\left\{x_{i},y_{i}\right\},\;i=1,2,\ldots 5,\;x_{i}$ is referred to as all data of the ith subset and $y_{i}$ is the label of the corresponding subset. In the first round, the union of $\left\{A_{2},A_{3},A_{4},A_{5}\right\}$ is used as the training set and $A_{1}$ is used as the validation set. Given the input $x_{1}$, AT-CNNs in this stage make corresponding predictions $p_{\mathrm{FZ}}\left(x_{1}\right),\;p_{\mathrm{FCZ}}\left(x_{1}\right)$,…$\;p_{n}\left(x_{1}\right)$, where $p_{n}\left(x_{1}\right)$ is a binary variable and $P_{n}\left(x_{1}\right)$ refers to the prediction result of the channel n. After the classifications in the first round, we assemble the predictions of each AT-CNN model into $P_{1}=[p_{\mathrm{FZ}}\left(x_{1}\right),\;p_{\mathrm{FCZ}}\left(x_{1}\right)$,…$\;p_{n}\left(x_{1}\right)]$, which is merged with the corresponding label y1 to form a new dataset $B_{1}$, for use in the second stage.

After five rounds, we obtain five new datasets, $\left\{B_{1},B_{2},B_{3},B_{4},\;B_{5}\;\right\},B_{i}=\left\{P_{i},y_{i}\right\}$, i=1,2,3,4,5. A simple three-layer neural network is used in the second stage as the ensemble model. The number of neurons in the input layer of the network is equal to the number of classifiers, and they represent the features of the samples in the new dataset. The number in the hidden layer is equal to 1/2 of the number of neurons in the input layer. The output layer of the network contains a neuron whose output is 0 or 1, indicating that ErrP is not detected and ErrP is detected, respectively.

Commonly used in the ensemble strategy is the majority voting algorithm [39]. The majority voting method does not consider the performance difference of each classifier. If a label receives more than half of the votes, it is predicted as the label. Due to the different prediction performance of each channel classifier, and the nonlinear relationship between each channel classifier and the sample label, the method of majority voting algorithm cannot guarantee the reliability of the prediction. In contrast, our method can automatically learn intricate relationships between different channels and requires very little engineering by hand.

## 3. Experiments and Results

### 3.1. Datasets

Dataset 1: The dataset used in this work was derived from the BNCI Horizon 2020 project website under the name Monitoring Error-Related Potential [15]. The experimental paradigm is shown in Figure 6. During the experiment, the user monitored the behaviors of the cursor, and the subjects were asked to expect the cursor to move toward the target. In order to obtain the ErrP signal, the experiment controlled the cursor with an error rate of 20%, that is, the cursor moved in the direction away from the target with the probability of 20%.

The dataset contains six participants, involving two sessions and a period of time between two sessions. Table 2 shows the time difference (in days) between the two sessions for all subjects. In the experiment, the EEG signals of all participants were recorded at a sampling rate of 512 Hz, and the acquisition process used 64 channels according to the 10–20 international standard.

Dataset 2: To further verify the effectiveness of the algorithm, we designed a new experiment. The dataset included eight subjects whose EEG signals were collected using the NeuSen W Series wireless EEG acquisition system manufactured by Neuracle. As shown in Figure 7, a white box appeared in the center of the screen. Two seconds later, a blue box appeared randomly to the left or right of the white box. The subjects were asked to expect the blue box to move to the position of the white box. The experimental control error rate was 30%, which meant that the box had a 30% chance of moving away from the white box. The experiment consisted of three blocks, with 30 trials in between each block and 10 min between each block.

As for the dataset segmentation of training and evaluation models, we simulated the practical application of the BCI system. In the development of practical BCI applications, the data of the subjects will be collected firstly and used to train the model. Then, the model will be applied to the future BCI system of the subjects Therefore, for dataset 1, the data from the first session are used for model training and validation, and the data from the second session are used for testing. For dataset 2, the first two blocks are used for model training and validation and the last block for the test set. The experimental results are the average of five repeated experiments. In order to better display the experimental results, we will combine and analyze the two datasets in the subsequent experiments. 

### 3.2. Performance of Different Channel Groups

We display and discuss the effect of choosing different channel groups on ErrP classification accuracy. ErrP features are generated in the ACC, in which features of channel FCZ are the most obvious, and the ErrP waveform features in the peripheral channels of channel FCZ gradually weaken. Therefore, the principle of channel selection is: in the region of ACC, with the FCZ as the center, selecting the channels around the FCZ. We considered five channel groups, and the spatial positions of the channels contained in each channel group are shown in Figure 8. The experimental results are shown in Table 3.

From Table 3, we can make some observations. Firstly, Group D (including channel F1, FZ, F2, FC1, FCZ, FC2, C1, CZ and C2) has the best overall performance (the accuracy difference to the control is statistically significant: *p* < 0.05), of which the accuracy is 86.46%, sensitivity is 72.46% and specificity is 90.17%. Secondly, from Group A-D we found that as the number of selected channels increases, the performance of the system increases (the accuracy increases from 80.85% to 86.46%, specificity increases from 83.70 to 90.17% and sensitivity increases from 55.53% to 72.46%). Thirdly, not all channel classifiers are helpful for the final integrated performance improvement with the increasing of channel group selection. This may be attributed to the fact that some channel classifiers are useless interference information for the ensemble model. 

### 3.3. Contribution of Different Techniques

The classification performance of the single-channel 2D EEG image may directly affect the performance of the multi-channel ensemble. To evaluate the contribution of different elements of the AT-CNN on system performance, we focused on Group D and tested our proposed AT-CNN when some elements were omitted. We employed the same hyperparameters (learning rate, batch size, etc.) for a fair comparison. Table 4 shows the experimental results. 

To test the contribution of label smoothing, we used cross-entropy as the loss function. We found that the accuracy dropped from 86.46% to 82.71%, and sensitivity dropped significantly from 72.46% to 46.45%.

Similarly, we trained AT-CNN again; when the model lacked BN layers, CBAM, and did not use the sample balancing training data, the accuracy dropped to 81.88%, 83.12% and 83.64%, respectively.

The experimental results show that the AT-CNN proposed in this paper can achieve good classification performance.

### 3.4. Multi-Channel Ensemble Based on Neural Networks

We integrated all the first stage predictions using a neural network. We first trained nine different channel classifiers of group D, and then used five-fold cross-validation to obtain the average performance. To verify the effectiveness of the proposed ensemble method, we compared the majority voting ensemble method with our proposed ensemble method.

The experimental results are shown in Table 5. It is clear that the performance of integrated multiple channel classifiers is better than a single-channel classifier, and our proposed ensemble method performs better, with an accuracy rate of 86.46% and majority vote method of 81.29%. The specificity of the majority voting method is higher than the method proposed in this paper, but the sensitivity is only 46.65%, because the majority voting method tends to predict the negative class. Our proposed ensemble method outperformed the majority voting method by 5.27% and achieved a better balance between specificity and sensitivity with the specificity of 90.17% and sensitivity of 72.46%; there is a significant difference between model accuracies (*p* < 0.05)

We also noticed that the performance of each channel classifier varies greatly. Classifier FCZ’s accuracy ranked first with 80.42% and classifier C1’s accuracy ranked last with 66.08%. Each channel classifier tends to predict negative class, although we adopt an oversampling strategy to increase positive samples.

### 3.5. Comparison with the Other Methods

In order to evaluate the proposed method, we compared the experimental results with other papers, including the dataset contributor Chavarriaga et al. [16], the LDA algorithm [18], the ConvNet algorithm [19], the CNN-based double transfer learning approach [20] and the MLP ANN [21]. To verify the advantages of 2D image classification over one-dimensional signal classification, we added a reference model named AT-CNNs-1D. AT-CNNs-1D have network architecture very similar to AT-CNNs-2D, while the dimension of the filter in the first stage is 1D, and the corresponding input signals are 1D EEG signals.

The data of session1 are used as the training set and the validation set, and the data of session 2 are used as the test set. The experimental results are shown in Table 6. The accuracy of our method is 86.46%, the sensitivity is 72.46% and the specificity is 90.17%. Our method achieved better results than other methods. There is a statistical significance (*p* < 0.05) between our model and the best previous model (MLP-ANN), indicating that the best model is AT-CNNs-2D.

## 4. Discussion

Based on the results, it can be observed that the proposed multi-channel ensemble method for ErrP classification outperforms the other methods. 

An important reason for the achievements of our method is that we integrate information from multiple channels of EEG signal. We considered five different channel groups of channels, all of which were selected from the anterior cingulate cortex (ACC), an area of the brain thought to be responsible for ErrP generation. From Table 3, we find that channel group D (containing nine channels) has the highest classification accuracy of 86.46%, although the number of channel group D is not the largest. It can be concluded that channel group D performs better than group A (containing two channels), group B (containing three channels) and group C (containing five channels), probably because the newly added channels can provide helpful local information, thereby improving the performance of the entire ErrP classification. The performance of channel group E (containing 15 channels) is not as good as that of channel group D, indicating that as the number of selected channels increases, the overall performance of the model does not always increase. We speculate that the reason may be that the newly added channels, such as channel F3, C3, etc., do not have enough discriminative features, resulting in poor performance of the AT-CNN model responsible for the decision of the channel. The local decision provided is not accurate enough, which increases the noise of the second stage input and then reduces the performance of the ensemble model. 

An important component of our proposed system is AT-CNN. We conducted some experiments to test the contribution of different elements of AT-CNN and different training settings. From Table 4, we found that the different elements of AT-CNN and different training settings we applied in this paper improved the performance of our system. The use of BN can effectively avoid model overfitting, thereby improving the performance of the system. In addition, using oversampling to balance the data can improve the sensitivity of the system. The convolutional block attention module (CBAM) in AT-CNN is very important for the performance of the entire system. The main reason is that spatial attention can help the model to better learn the spatial locality features of 2D EEG image, so that the model can focus on areas in the image that are easy to distinguish ErrP and no-ErrP. Channel attention can help the model to better learn the relationship among each feature channel. In addition, our proposed system uses label smoothing during training to avoid the model fitting the data to extremes. Label smoothing is also helpful for system performance improvement. Compared with using cross-entropy, after using label smoothing in the training process, the model performance is better. The main reason is that some labels in the training data do not match the actual ones and label smoothing improves the generalization ability of the model. 

Effective integration of prediction information from multiple channels is critical to improve the performance of ErrP classification. We proposed a deep learning-based ensemble strategy to integrate the decisions of different channel classifiers in the first stage.

From Table 5, we find that the performance of different channel classifiers varies widely. The highest classification accuracy is channel FCZ, reaching 80.42%, and the worst is channel C1, which is only 66.08%. We compared the two ensemble methods; the majority voting method has a classification accuracy of 81.29%, and our proposed ensemble method has a classification accuracy of 86.46%. The accuracy of the ensemble method is significantly higher than the classifier of a certain channel, because the ErrP dataset is not large enough due to the experimental design and practical application scenarios. When only a small amount of data is available, it is easy for machine learning algorithms to find different hypotheses that fit the training data perfectly, but perform poorly on the test data, whereas our hypothesis of integrating multiple channel classifiers can effectively reduce the risk of choosing the wrong hypothesis, thereby improving overall forecasting performance.

The classification accuracy of our proposed ensemble method improves by 5.27% over the majority voting method. It may be that majority voting only considers the linear relationship without considering the weight of each classifier. As for the characteristics of the EEG signal itself, the signals between each channel have a certain relationship, so the relationship between each classifier is not completely independent. Compared with the majority voting, the proposed method uses a neural network to integrate the outputs of different channel classifiers, which can automatically learn the complex relationships among the different channel classifiers, leading to better and more stable performance.

The specificity of the majority voting method is higher than that of the method proposed in this paper, but the sensitivity is only 46.65% because a single-channel classifier tends to predict the negative class. Obviously, the performance of the majority voting method is poor to some extent because the relationship between different channels and labels is not a simple linear relationship. The method proposed in this paper achieves a better balance between specificity and sensitivity, which proves that our proposed method can learn the nonlinear relationship between multiple channels and labels, such as learning which channels are more useful for classification. In addition, our method can make full use of the information provided by the original data and reduce hand engineering.

The comparison with other methods is shown in Table 6. Firstly, traditional methods, such as Gaussian classifier [15] and LDA [18] perform worse than deep learning-based methods (ConvNet [19], paper [20], MLP-ANN [21] and ours). This indicates that deep learning-based methods can learn from the data itself and learn more discriminative features. The method in paper [20] is better than ConvNet [19] and MLP-ANN [21], showing that classification performance can be improved by using other subjects to learn global features, and then finetuning model parameters to adapt to each subject. Secondly, the accuracy of AT-CNNs-1D was 83.18% lower than that of AT-CNNs-2D (86.46%). The result indicates that converting one-dimensional EEG signals into 2D waveform images is more helpful to improve classification performance; for this purpose, 2D CNN is less sensitive to noise than 1D CNN. Two-dimensional CNN is more conducive to capturing the microstructural detail of the input data.

Thirdly, we can find that the sensitivity of all methods is lower than accuracy and specificity, and the fluctuation range is relatively large. On the one hand, there are a few samples containing wrong samples, so the positive and negative samples are unbalanced, and the model tends to predict positive samples. On the other hand, the signal-to-noise ratio of the EEG signal itself is low, so the ErrP feature is latent in the EEG signal, which has a great impact on the recognition accuracy. Overall, the accuracy of our proposed method is 86.46%, the sensitivity is 72.46% and the specificity is 90.17%, and our method achieves the best classification performance compared to other methods. That is to say, it is more advantageous to use 2D EEG images to detect ErrP. From the perspective of stability, our method has the best stability, because we combine the decision results of multiple channels, which reduces the variance and obtains more robust results.

## 5. Conclusions and Future Work

In this paper, we proposed an effective multi-channel ensemble method for ErrP classification method using 2D EEG images. We used 2D EEG images as input data after signal preprocessing and the effect of the noise in the signal was minimized. A deep convolution network model named AT-CNN was proposed to effectively extract the spatiotemporal features of 2D EEG images. The multi-channel ensemble approach then takes predictions from multiple different channels as input to an ensemble model, which is trained to combine multiple channel predictions to form a final prediction. We analyzed the prediction results of multiple channels and compared the majority voting method with our proposed multi-model ensemble method. The results show that the accuracy of our proposed ensemble model outperforms other every channel classifier as well as the majority voting method. Finally, we compared the proposed method with other methods; the results demonstrate that our method has the best performance.

In terms of future work, we will aim to investigate transfer learning in the application of ErrP. If the time interval between two sessions was too long, the ErrP characteristics of the subjects would change greatly, thus reducing the detection accuracy. Transfer learning provides a solution to eliminate the differences between the two sessions and improve the detection performance of the model.

## Figures and Tables

**Figure 1 sensors-23-02863-f001:**
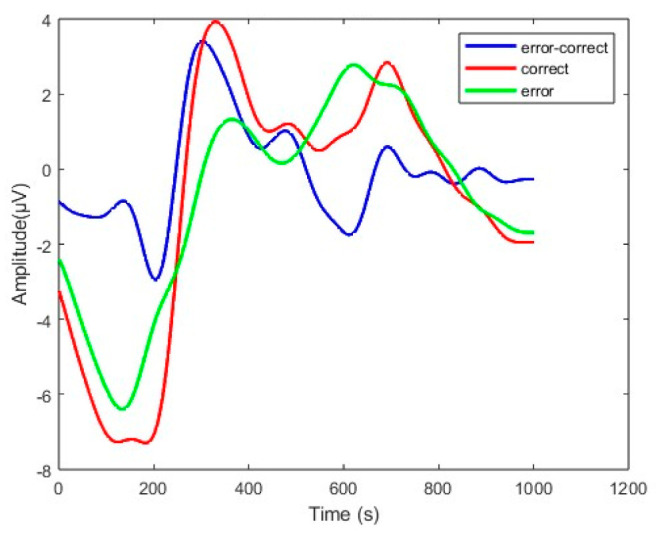
Grand average ErrP at channel FCz for correct, erroneous and different (error minus correct) conditions. t = 0 corresponds to the stimulus presentation onset.

**Figure 2 sensors-23-02863-f002:**
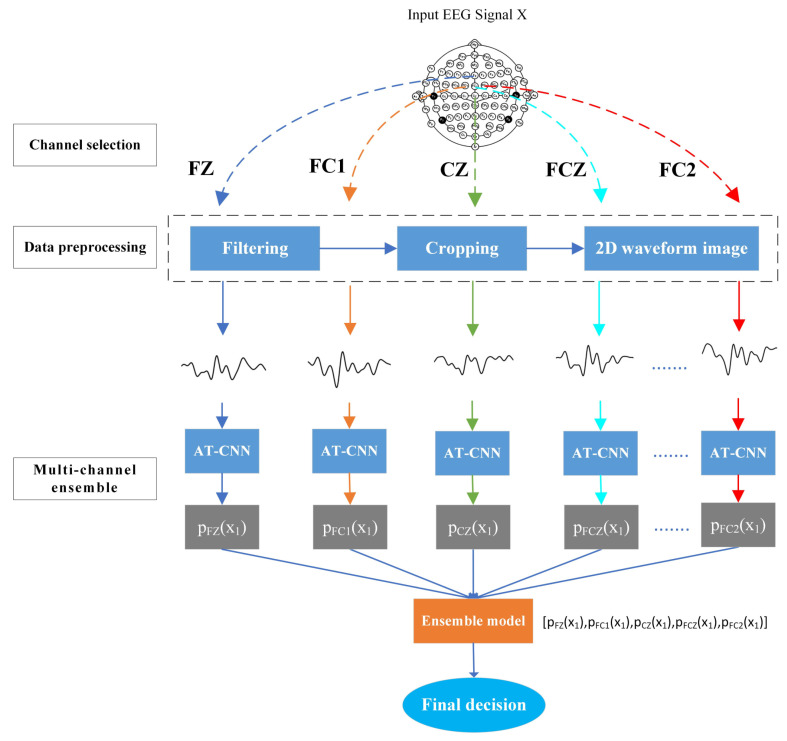
The overview of the proposed multi-channel ensemble method for ErrP detection using 2D EEG images.

**Figure 3 sensors-23-02863-f003:**
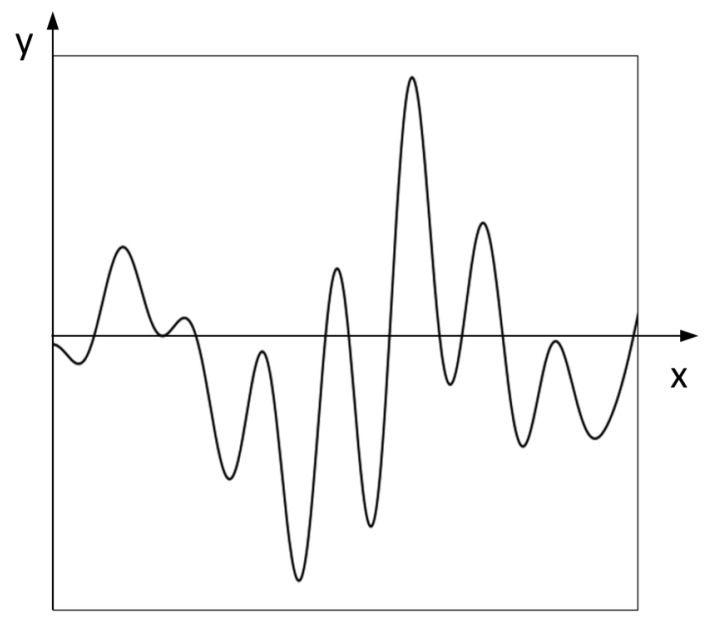
ErrP elicited at channel FCZ for a signal trail. (The x and y axes are shown for the sake of illustration, but actually, the x and y axes are hidden).

**Figure 4 sensors-23-02863-f004:**
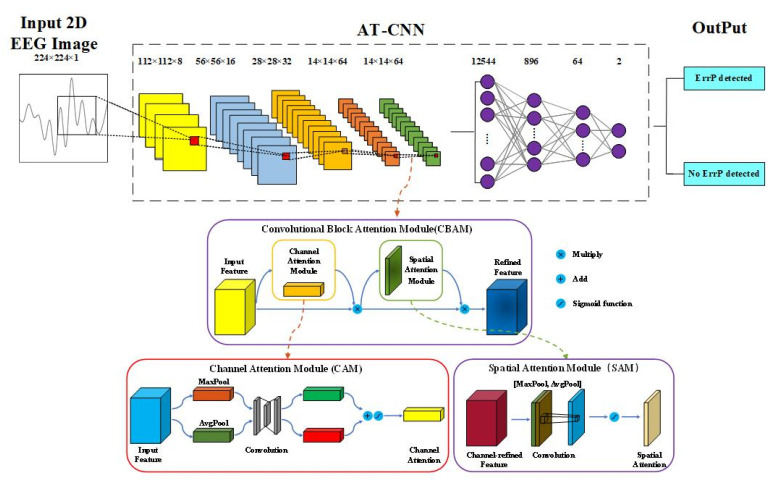
The schematic diagram of AT-CNN identifying a 2D EEG Image. Inside the dotted line is the proposed attention-based convolutional neural network. The convolutional block attention module (CMAM) is added before the fully connected layer and it contains a channel attention module (CAM) and a spatial attention module (SAM).

**Figure 5 sensors-23-02863-f005:**
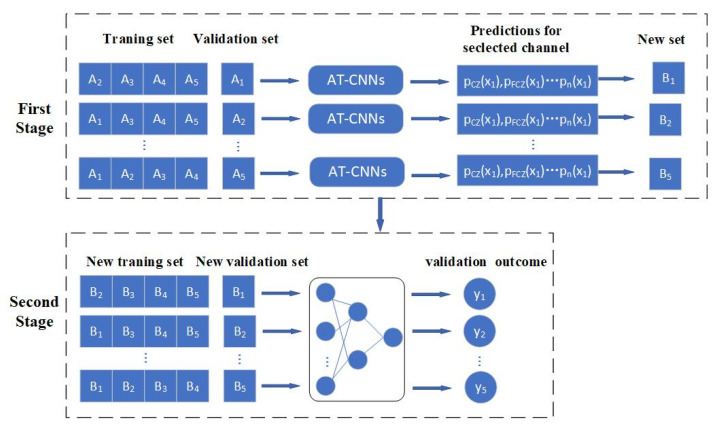
The training process of AT-CNN of each channel and the ensemble model.

**Figure 6 sensors-23-02863-f006:**
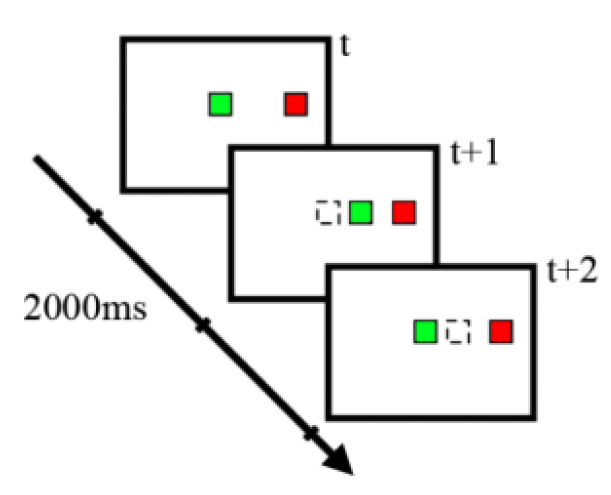
Experimental paradigm. Two squares are displayed on the horizontal line in the center of the screen; one is the cursor (green square), the other is the target (red square), and the dotted square is the previous position of the cursor. The interval time of cursor movement is 2 s, and the cursor moves away from the target with the probability of 20%.

**Figure 7 sensors-23-02863-f007:**
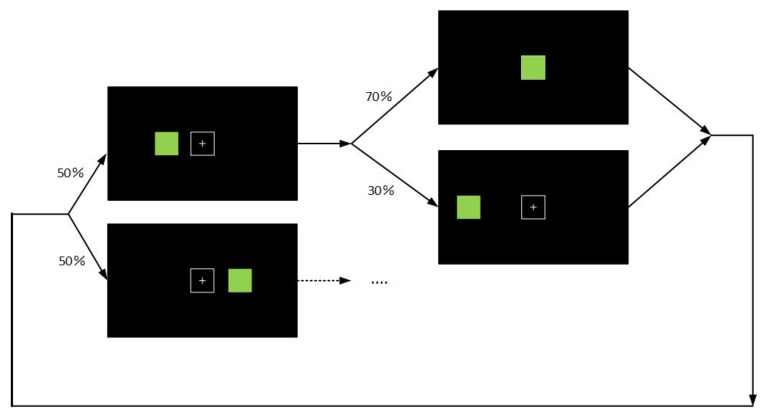
Experimental paradigm of dataset 2. A green square and a white box are displayed on the horizontal line in the center of the screen. With the same probability, the green square appears either on the left side or on the right side of the white box. After a 4 s interval, the green square moves, and there is a probability of 70% that the green square will move to the white box, and a probability of 30% that it will move away from the white box.

**Figure 8 sensors-23-02863-f008:**
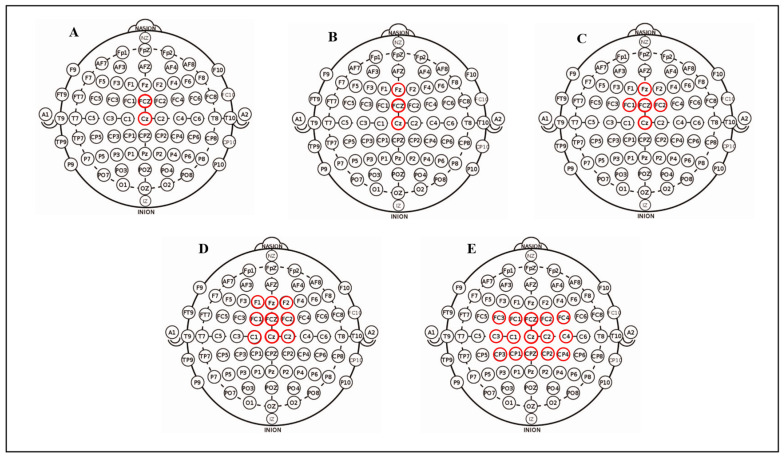
Visualization of the five channel groups chosen in the regions of anterior cingulate cortex. The five channel groups (**A**–**E**) illustrate the position of the 64 EEG electrodes on the scalp (small circles, each reporting the standard designation). The channels forming the channel group are highlighted in red.

**Table 1 sensors-23-02863-t001:** Architecture of AT-CNN.

Module/Features	Type	Input Size	Filters	Kernel Size	Stride	Activation
M1	2D Convolution	(224,224,1)	8	(3,3)	1	ReLu
	Batch Normalization	(224,224,8)				
	Max Polling	(224,224,8)		(2,2)	2	
M2	2D Convolution	(112,112,8)	16	(3,3)	1	ReLu
	Batch Normalization	(112,112,16)				
	Max Polling	(112,112,16)		(2,2)	2	
M3	2D Convolution	(56,56,16)	32	(3,3)	1	ReLu
	Batch Normalization	(56,56,32)				
	Max Polling	(56,56,32)		(2,2)	2	
M4	2D Convolution	(28,28,32)	64	(3,3)	1	ReLu
	Batch Normalization	(28,28,64)				
	Max Polling	(14,14,64)		(2,2)	2	
M5	CMAM	(14,14,64)				
M5	Flatten	(1,12544)				
M6	Fully-connected	(1,12544)				ReLu
	Dropout	(1,12544)				
M7	Fully-connected	(1,896)				ReLu
	Dropout	(1,896)				
M8	Fully-connected	(1,64)				Softmax
	Dropout	(1,64)				

**Table 2 sensors-23-02863-t002:** Time difference (in days) between the two sessions.

Subject	1	2	3	4	5	6
Days between two sessions	51	50	54	211	628	643

**Table 3 sensors-23-02863-t003:** Performance results of different channel groups.

Metrics\Group	A	B	C	D	E
Accuracy (%)	80.85 ± 2.12	82.36 ± 1.14	83.20 ± 1.63	86.46 ± 1.87	83.59 ± 1.51
Specificity (%)	83.70 ± 2.36	84.52 ± 6.13	85.08 ± 3.15	90.17 ± 1.12	86.23 ± 1.41
Sensitivity (%)	55.53 ± 4.61	56.38 ± 5.94	64.61 ± 4.51	72.46 ± 3.45	66.35 ± 2.67

**Table 4 sensors-23-02863-t004:** Performance of AT-CNN when some elements are omitted.

Metrics\Technique	No Label Smoothing	No BN	No CBAM	No Oversampling	FCZ
Accuracy (%)	82.71	81.88	83.12	83.64	86.46
Specificity (%)	85.28	87.30	86.58	91.06	90.17
Sensitivity (%)	46.45	57.21	61.93	45.97	72.46

**Table 5 sensors-23-02863-t005:** The performance indices (%) of individual channel and ensemble methods.

Metrics\Classifiers	F1	FZ	F2	FC1	FCZ	FC2	C1	CZ	C2	Majority Voting	Ours
Accuracy (%)	69.19	74.81	76.85	78.61	80.42	77.12	66.08	80.18	77.72	81.29 ± 2.13	86.46 ± 1.87
Specificity (%)	72.57	80.15	81.38	85.40	83.48	84.12	67.45	84.78	83.51	91.20 ± 2.41	90.17 ± 1.12
Sensitivity (%)	55.65	53.45	30.72	45.45	58.17	49.12	60.61	56.78	54.54	46.65 ± 1.57	72.46 ± 3.45

**Table 6 sensors-23-02863-t006:** Performance indices of the different methods.

Methods\Metrics	Accuracy (%)	Sensitivity (%)	Specificity (%)
Gaussian classifier [16]	70.41 ± 7.40	63.21 ± 9.06	75.81 ± 6.84
LDA [18]	68.41 ± 5.43	63.15 ± 7.46	70.40 ± 5.95
ConvNet [19]	71.39 ± 2.81	55.17 ± 3.84	76.92 ± 3.64
Paper [20]	75.56 ± 3.46	64.43 ± 4.45	80.14 ± 2.60
MLP-ANN [21]	82.17 ± 2.17	59.43 ± 1.31	85.61 ± 1.96
AT-CNNs-1D	83.18 ± 1.32	61.91 ± 3.17	86.70 ± 1.43
AT-CNNs-2D (proposed)	86.46 ± 1.87	72.46. ± 3.45	90.17 ± 1.12

## Data Availability

Monitoring Error-Related Potential dataset at http://bnci-horizon2020.eu/database/data-sets (URL (accessed on 1 September 2022)).

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
