# Peer review of "A Multi-Channel Ensemble Method for Error-Related Potential Classification Using 2D EEG Images"

_sensors, 2023, doi:10.3390/s23052863_

Round 1

Reviewer 1 Report

In this study, a multi-channel ensemble approach to effectively integrate channel classifier decisions which ensemble approach learns the nonlinear relationship between each channel and the label, achieving 2.47% higher accuracy than the majority voting ensemble approach which had 86.76% accuracy, 69.02% sensitivity, and 91.17% specificity. This paper's AT-CNNs-2D improve ErrP classification accuracy and offer new brain-computer interface classification ideas. Even though the results and the inferences of this paper are publication worthy, the authors could consider the following suggestions to further improve the quality of the article:

1)      The size and the clarity of the figure 1, 2 and 3 is not upto publication standards.

2)      Lines 53 and 54 must be supported with reference.

3)      One of the recent developments in machine learning is the usage of snapshot based hyperspectral imaging combined with machine learning. Even though the authors have discussed in detail on the literature review, the authors have failed to mention about the usage of HSI. Some examples are given below:

a.      Kai-Chun Li, et al.” Intelligent Identification of MoS2 Nanostructures with Hyperspectral Imaging by 3D-CNN,” Nanomaterials 10(6), 1161 (2020).

b.      C.-W. Chen, et al. “Air Pollution: Sensitive Detection of PM2.5 and PM10 Concentration Using Hyperspectral Imaging,” Appl. Sci. 2021, 11, 4543 (2021).

c.       Chen, Chi-Wen, et al."Air Pollution: Sensitive Detection of PM2. 5 and PM10 Concentration Using Hyperspectral Imaging." Applied Sciences 11, no. 10 (2021): 4543.

4)      Figure 8 and table 3 shows the same information/results in two different formats. Either the table or the figure could be moved to the supplementary file.

Author Response

Point 1: The sample size of subjects (N=6) is too small in the selected public dataset. The authors are suggested to use some other datasets for further evaluation of the performance.

Response 1: Thanks for your suggestion. According to your suggestion, we designed a new paradigm and collected the data of 8 subjects for the experimental analysis.

Point 2: Statistical analysis is missed in the current manuscript.  Is there significant difference between AT-CNN and existed algorithms?

Response 2: Thanks for your suggestion, we have added T-test to the new manuscript.

Point 3: As spatiotemporal features were used for classification in this work, the ocular artifacts and other low-frequency artifacts may have strong effects on the classifications.  So, the authors are suggested to have a detail description of the preprocess, especially the methods for artifacts rejection.

Response 3: Thank you for your advice.The raw EEG signal contains a lot of noise. To improve the signal-to-noise ratio, the raw EEG signal is spatially filtered using common average reference (CAR) and then filtered to [1-10] Hz using a third-order Butterworth band-pass filter. To eliminate eye artifacts and other low-frequency artifacts, we use ICA for artifact elimination. The preprocessing description has been added to the new manuscript.

Point 4: For the 2D EEG images, was the vertical axis aligned?  And what was the y-axis scale?

Response 4: Thank you for your suggestion. This is our problem that we did not state it in the original manuscript clearly. For 2D EEG images, the vertical axis is aligned. As shown in Figure 3, we assume that the lower left corner of the image is the reference coordinate (0,0) and the origin of the coordinate is located at the position of coordinate (0,124). The x axis is the direction of the time axis, and the y axis is the direction of the amplitude. We normalize the maximum and minimum values of the signal, that is, the signal amplitude is limited to between [0,1], so the scale on the Y-axis is 1/124.

Point 5:The authors have selected five channel groups for comparison of decoding results.  However, the channel locations and channel numbers were both changed among these groups.  So, how to explain the relationship between these two factors (channel number, channel location) and the decoding accuracy?

Response 4: The signal features of ErrP appear in the anterior cingulate cortex, especially in channel FCZ, and the signal features show a decreasing trend around channel FCZ. Therefore, in groupA-D, with the expansion of the selection range of the channel centered on FCZ (the number of channels increases), our method can fuse the signal features of multiple channels to improve the performance of the overall system. However, with the expansion of the channel selection range, the newly added channels no longer provide valuable information, but will introduce noise. For channel group E, We speculate that the reason may be that the newly added channels such as channel F3 and C3, etc.,  do not have enough discriminative features, resulting in poor performance of the AT-CNN model responsible for the decision of the channel. The local de-cision provided is not accurate enough,  which increases the noise of the second stage input and then reduces the performance of the ensemble model.

Reviewer 2 Report

The authors proposed a model named attention-based convolutional neural network (AT-CNN) for the classification of ErrP. In addition, a multi-channel ensemble approach was proposed for integrating the multi-channel decisions. Finally, the performance of the algorithm was evaluated on a public dataset. However, the performance improvement was not significant compared with the existed methods. Some critical issues need to be addressed to improve the quality of the manuscript.

 1.     The sample size of subjects (N=6) is too small in the selected public dataset. The authors are suggested to use some other datasets for further evaluation of the performance.

2.     Statistical analysis is missed in the current manuscript. Is there significant difference between AT-CNN and existed algorithms?

3.     As spatiotemporal features were used for classification in this work, the ocular artifacts and other low-frequency artifacts may have strong effects on the classifications. So, the authors are suggested to have a detail description of the preprocess, especially the methods for artifacts rejection.

4.     For the 2D EEG images, was the vertical axis aligned? And what was the y-axis scale?

5.     The authors have selected five channel groups for comparison of decoding results. However, the channel locations and channel numbers were both changed among these groups. So, how to explain the relationship between these two factors (channel number, channel location) and the decoding accuracy?

Author Response

Point 1: The size and the clarity of the figure 1, 2 and 3 is not upto publication standards

Response 1: Thanks for your suggestion and we have updated these images.

Point 2: Lines 53 and 54 must be supported with reference.

Response 2: Thanks for your suggestion, we have added the references.

Point 3:   One of the recent developments in machine learning is the usage of snapshot based hyperspectral imaging combined with machine learning. Even though the authors have discussed in detail on the literature review, the authors have failed to mention about the usage of HSI. Some examples are given below:

  1. Kai-Chun Li, et al.” Intelligent Identification of MoS2 Nanostructures with Hyperspectral Imaging by 3D-CNN,” Nanomaterials 10(6), 1161 (2020).
  2. C.-W. Chen, et al. “Air Pollution: Sensitive Detection of PM2.5 and PM10 Concentration Using Hyperspectral Imaging,” Appl. Sci. 2021, 11, 4543 (2021).
  3. Chen, Chi-Wen, et al."Air Pollution: Sensitive Detection of PM2. 5 and PM10 Concentration Using Hyperspectral Imaging." Applied Sciences 11, no. 10 (2021): 4543.

Response 3: Thank you for your suggestion and we have incorporated the recommended references into the manuscript.

Point 4: Figure 8 and table 3 shows the same information/results in two different formats. Either the table or the figure could be moved to the supplementary file.

Response 4: Thank you for your suggestion. In order to better present the experimental results, we have simplified the table

Round 2

Reviewer 1 Report

The authors reply all my comments. This article can be accepted by Sensors.

Reviewer 2 Report

1. Please add the x and y axis range in Figure 3. It will help the authors to understand the figure features.

2. Is there any interval between different trials in your own experiment? How long is the interval?